# An Assessment of Global Macroeconomic Impacts Caused by Sea Level Rise Using the Framework of Shared Socioeconomic Pathways and Representative Concentration Pathways

**Osamu Nishiura [1,*], Makoto Tamura [2], Shinichiro Fujimori [1,3,4] , Kiyoshi Takahashi [3] , Junya Takakura [3] and Yasuaki Hijioka [3]**

[1]    Department of Environmental Engineering, Kyoto University, C1-3, Kyotodaigaku-Katsura, Nishikyo-ku, Kyoto 615-8540, Japan; fujimori.shinichiro.8a@kyoto-u.ac.jp

[2]    Global and Local Environment Co-creation Institute, Ibaraki University, 2-1-1 Bunkyo, Mito, Ibaraki 310-8512, Japan; makoto.tamura.rks@vc.ibaraki.ac.jp

[3]    National Institute for Environmental Studies, 16-2 Onogawa, Tsukuba, Ibaraki 305-8506, Japan; ktakaha@nies.go.jp (K.T.); takakura.junya@nies.go.jp (J.T.); hijioka@nies.go.jp (Y.H.)

[4]    International Institute for Applied Systems Analysis, Schlossplatz-1, 2361 Laxenburg, Austria

[*]    Correspondence: nishiura.osamu.56s@st.kyoto-u.ac.jp

**Abstract:** Coastal areas provide important services and functions for social and economic activities. Damage due to sea level rise (SLR) is one of the serious problems anticipated and caused by climate change. In this study, we assess the global economic impact of inundation due to SLR by using a computable general equilibrium (CGE) model that incorporates detailed coastal damage information. The scenario analysis considers multiple general circulation models, socioeconomic assumptions, and stringency of climate change mitigation measures. We found that the global household consumption loss proportion will be 0.045%, with a range of 0.027−0.066%, in 2100. Socioeconomic assumptions cause a difference in the loss proportion of up to 0.035% without greenhouse gas (GHG) emissions mitigation, the so-called baseline scenarios. The range of the loss proportion among GHG emission scenarios is smaller than the differences among the socioeconomic assumptions. We also observed large regional variations and, in particular, the consumption losses in low-income countries are, relatively speaking, larger than those in high-income countries. These results indicate that, even if we succeed in stabilizing the global mean temperature increase below 2 °C, economic losses caused by SLR will inevitably happen to some extent, which may imply that keeping the global mean temperature increase below 1.5 °C would be worthwhile to consider.

**Keywords:** climate change; sea level rise; CGE model; RCP/SSP scenario

## 1. Introduction

In recent years, climate change has become a great concern worldwide. At the 21st Conference of the Parties of the United Nations Framework Convention on Climate Change, the Paris agreement was adopted as a new framework from 2020, when the second commitment period of the Kyoto Protocol ends [1]. As of February 2020, 197 countries have become parties to the Paris agreement and each country must determine its contribution to mitigating climate change and report its progress on this to the United Nations Framework Convention on Climate Change every five years. The Paris agreement aimed not only to mitigate climate change but also to adapt to its adverse impacts. In the Paris agreement, the importance of support for developing countries that are especially vulnerable to climate change is emphasized [2].

Low-lying coastal areas are among the main places threatened by exposure to climate change. Sea level rise (SLR) has already been observed and is projected to go further due to climate change. According to the fifth assessment report of the Intergovernmental Panel on Climate Change (IPCC), the global mean sea level has risen by 0.19 m from the level of 1910 [3]. Under all calculated scenarios, the rate of SLR will very likely exceed that observed during 1971 to 2010 [4]. According to McGranahan et al. [5], coastal areas provide important services and functions for human activities, and many major settlements have been developed there. The low-elevation coastal zone constitutes 2% of the world's land area, but 10% of its population and about 65% of its cities with populations greater than five million are located in this zone. For many locations, coastward migration, coastal industrialization, and urbanization grow this zone's population and assets faster than the national average trends [5]. Because of the geographical and socioeconomic conditions in particular low-income countries, the damage caused by SLR is considered a major threat among climate change impacts, and one that can harm their economic growth. Therefore, an economic assessment of SLR and adaptation measures for it would be meaningful for policymakers.

Many assessments of the global scale of coastal impacts, up to now, have been conducted with the Dynamic and Interactive Coastal Vulnerability Assessment (DIVA) model. Bosello et al. [6] and Pycroft et al. [7] introduced the data from the DIVA model to a computable general equilibrium (CGE) model to consider its effect on that general equilibrium. They revealed that economic assessment using only direct damage can be underestimated and emphasized the need for general equilibrium effects to be considered. However, in their research, it was difficult to assess the impacts of climate change mitigation under different scenarios. Schinko et al. [8] also used CGE and growth models to assess the macroeconomic impacts of SLR based on representative concentration pathways (RCPs) and shared socioeconomic pathways (SSPs), scenarios that have been widely used to assess the impact of climate change since the time of the Intergovernmental Panel on Climate Change's fifth assessment report. They used the data of direct impact due to SLR from Hinkel et al. [9], which considered not only SLR but also coastal extreme water levels due to surges and tides. They projected that the global economic losses could be more than 3% in RCP2.6 and more than 4% in RCP4.5 without further adaptation and assuming high ice-melting. They used flood damage data, which was calculated by the DIVA model considering the probability density of extreme water levels, but the uncertainty of this damage is great after a 100 year period. Therefore, this uncertainty can affect the economic model calculation.

Previous studies have limitations in assessing the climate change mitigation impact, comparing among related studies and with the uncertainty of extreme sea levels. The main purpose of this study was to assess the global economic impact of SLR without extreme sea levels, including the indirect effect, based on the RCP and SSP scenarios. We have used a global general equilibrium model, named the Asia-Pacific Integrated Model/Hub (AIM/Hub, formerly named AIM/CGE) to assess the macroeconomic impacts caused by SLR based on Tsuchida et al. [10] and Tamura et al. [11], which evaluate the impact of inundation by SLR on the coastal zone, without extreme water levels. We calculate the change of household consumption as a welfare measure and identify regions and situations where the changes are relatively large. Tsuchida et al. [10] and Tamura et al. [11] combine the four RCP scenarios (RCP2.6, RCP4.5, RCP6.0, and RCP8.5) and the three SSP scenarios (SSP1, SSP2, and SSP3) to allow for the assessment of their economic impact, taking into account the uncertainties of the future social situation and climate condition. By setting the social situation based on the SSP scenarios when calculating with the CGE model, the change in the equilibrium effect due to the changed social situation is considered. Besides, using the RCP and SSP scenarios makes it easier to compare with other new studies.

This paper proceeds as follows. Section 2 presents the methods and new features of Tsuchida et al. [10] and Tamura et al. [11], and our method for assessing the macroeconomic impact of SLR. The results in Section 3 and the discussion in Section 4 focus on changes in household consumption from business-as-usual scenarios, which do not consider the impact of SLR, as a welfare measure.

## 2. Materials and Methods

### 2.1. Overview

In this research, we assess the macroeconomic impact of SLR using the AIM/Hub model based on the data from Tsuchida et al. [10] and Tamura et al. [11]. The target period is from 2005 (base year) to 2100. The target region is the whole world, divided into following 17 regions grouped by geography and economy: USA (USA), EU (XE25), rest of EU (XER), Turkey (TUR), Australia and New Zealand (XOC), China (CHN), India (IND), Japan (JPN), rest of East and South East Asia (XSE), rest of Asia (XSA), Canada (CAN), Brazil (BRA), rest of South America (XLM), former USSR (CIS), Middle East (XME), North Africa (XNF), and Sub-Sahara (XAF). To comprehensively understand the uncertainties of the global circulation models (GCMs) and future climate change mitigation policies, the SLR is calculated using four GCMs (MIROC-ESM [12], GFDL-ESM2M [13], NorESM1-M [14], and IPSL-CM5A [15]) based on four RCP scenarios (RCP2.6, RCP4.5, RCP6.0, and RCP8.5) [16], and the economic impact is calculated based on three SSP scenario (SSP1, SSP2, and SSP3) [17]. The RCP scenarios are the greenhouse gas (GHG) concentration trajectories based on the future radiative forcing, and they describe different climate futures. The four scenarios used in this study mean that the future radiative forcing reach 2.6 W/m2, 4.5 W/m2, 6.0 W/m2, and 8.5 W/m2 in 2100, respectively. The IPCC's Fifth Assessment Report WG1 reports that the average temperature increase over the late-21st century (2081–2100) compared to the early-21st century (1986–2005) reaches 1.0 °C (0.3–1.7 °C) under RCP 2.6, 1.8 °C (1.1–2.6 °C) under RCP 4.5, 2.2 °C (1.4–3.1 °C) under RCP 6.0, and 3.7 °C (2.6–4.8 °C) under RCP 8.5. The SSP scenarios are based on two dimensions, the level of challenges to adaptation and that of mitigation. SSP1, SSP2, and SSP3 used in this study assume a "sustainability" scenario that is easy to adapt and mitigate, a "middle of the road" scenario that continues the current trend, and a "regional rivalry" scenario that is difficult to adapt and mitigate, respectively. Under SSP2, the world population reaches 8.9 billion, and the world GDP reaches $290 trillion in 2100. SSP1 assumes a society in which population growth stops (6.8 billion) because of the assumption that emission reductions are easy to achieve. In addition, a society with a large GDP ($310 trillion) is assumed because of the assumption that it is easy to pay the costs of adaptation measures. On the contrary, SSP3 assumes a society with a growing population (12 billion) and relatively small GDP growth ($150 trillion). The combination of RCP and SSP scenarios allows the assessment of climate change impacts to take into account the uncertainty of the future socioeconomic condition and the stringency of mitigation policy.

Figure 1 shows an overview of the estimation method and Table 1 shows the input data in each process. The following describes the overall research process. We begin with the various GCMs providing information about future climate change, which is translated into the area of the coastal zone that is inundated and its impact based on the RCP scenarios [10,11]. The inundation area is calculated by adding the tidal change to the SLR and comparing this with topographic data. The inundation impact, represented in economic terms, is computed by a regression formula that is derived from historical disaster data in the EM-DAT database of the Centre for Research on the Epidemiology of Disasters [18]. Within this regression, gross domestic product (GDP), population, and inundation area are explanatory variables and they are, thus, used in the future estimate, where the population and GDP data are taken from the SSP scenarios and the above-mentioned inundation area is based on the climate model outcomes.

Then, the economic inundation impact is fed into the AIM/Hub model as a capital loss. In this model, economic value directly related to production activities (production machinery, factories, etc.) and social capital (roads, bay ports, etc.) are assumed as capital. The impact of SLR is calculated by subtracting the amount of capital loss due to inundation from the total capital within the CGE model. Note that non-market value, such as the loss of landscape and recreational value due to the loss of beach, cannot be dealt with via this method.

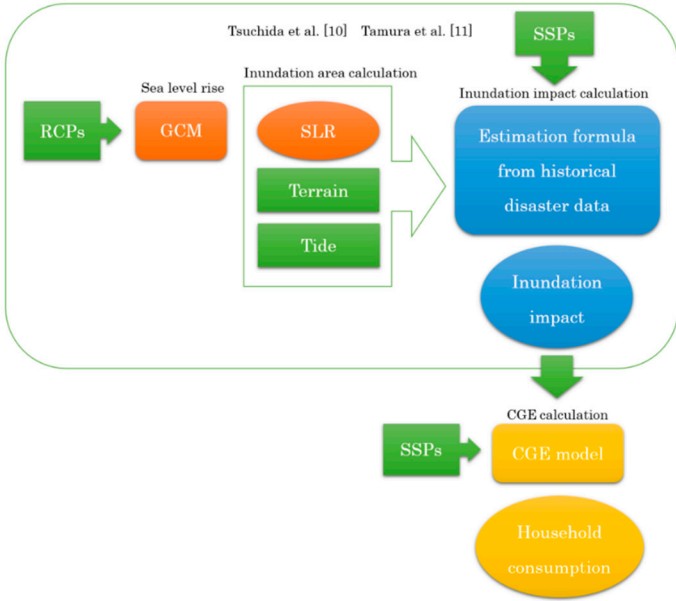

**Figure 1.** Overview of this study. Terms: representative concentration pathway (RCP), global circulation model (GCM), sea level rise (SLR), shared socioeconomic pathway (SSP), computable general equilibrium (CGE).

**Table 1.** Input data in each process. Terms: greenhouse gas (GHG), gross domestic product (GDP).

| Process | Input data | Source |
|---|---|---|
| (1) SLR calculation | GHG concentration | RCP scenarios [16] |
| (2) Inundation area calculation | Sea level<br>Land elevation<br>High tide water level | (1)<br>ETOPO1 [19]<br>TPXO7.2 [20] |
| (3) Multiple regression analysis | Historical disaster damage<br>GDP, Population | EM-DAT [18]<br>World Bank's statistics [21] |
| (4) Inundation impact calculation | Inundation area<br>GDP, Population | (2)<br>SSP scenarios [17] |
| (5) CGE calculation | Inundation impact<br>Socioeconomic conditions | (4)<br>SSP scenarios [17] |
| (6) Economic analysis | Household consumption | (5) |

*2.2. Impact Assessment of Inundation*

This section briefly describes the methods of Tsuchida et al. [10] and Tamura et al. [11]. The future monthly SLR under the RCP scenarios are taken from the GCM outcomes. Then, the tide is considered by adding the tidal change from the data of TPXO7.2 [20] to the SLR calculated by the GCM. The spatial distribution of the inundation area is calculated by comparing the topographic data based on ETOPO1 [19] with the SLR. The calculated inundation area and the distribution of population and GDP from the SSP scenarios are applied to the estimation formula to calculate the inundation impact. For more details, see Tsuchida et al. [10] and Tamura et al. [11].

Their results are shown in Figures 2 and 3. Figure 2 shows the cumulative inundation impact worldwide. The inundation impacts are large in SSP1 and small in SSP3. Figure 3 shows the inundation impacts per GDP by country in the year 2100 under RCP8.5 and under the three SSP scenarios. The developing regions XSA, XSE, XME, CHN, XOC, and XLM show relatively large inundation impacts.

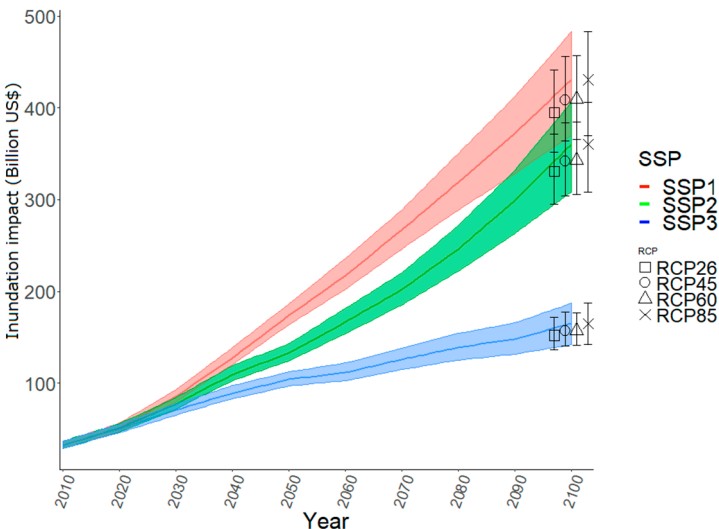

**Figure 2.** The cumulative inundation impact worldwide (billion US$). The lines show the average values of the results by four general circulation models and the ranges show the upper and lower limits of the results for these under representative concentration pathways (RCP) scenario number 8.5. The points show the average value and the error bars show the upper and lower limits in 2100 under each RCP scenario.

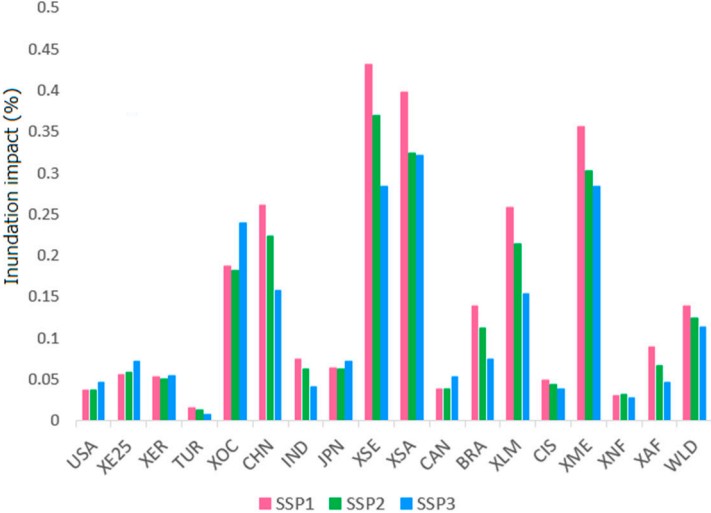

**Figure 3.** The cumulative inundation impacts divided by gross domestic product (GDP) in 2100 (%). The bar shows the average values of the results by four general circulation models under representative concentration pathways (RCP) scenario number 8.5.

## 2.3. CGE Model Calculation

The AIM/Hub model (formerly named AIM/CGE) has been used as a global CGE model (Fujimori et al. [22], Fujimori et al. [23], Takakura et al. [24], Hasegawa et al. [25]) to estimate the macroeconomic damage associated with SLR. The model has 43 industrial, government, household, and corporate sectors and finds an equilibrium solution every year from 2005 to 2100. Each sector has its capital stock, investing every year and depleting capital by 4% per year. The sectoral allocation of this investment is determined by the capital price and, once deployed, it is fixed for that sector. The production sectors maximize profits under multi-nested constant elasticity substitution (CES) functions and individual input prices. Household expenditures on each commodity are described by a linear expenditure system (LES) function. The parameters adopted in the LES function are recursively updated in accordance with income elasticity assumptions. The savings ratio is endogenously

determined to balance savings and investment, and capital formation for each good is determined by a fixed coefficient. The Armington assumption is used for trade (CES and constant elasticity of transformation function is used), and the current account is assumed to be balanced. For details, see Fujimori et al. [22].

Tsuchida et al. [10] and Tamura et al. [11] calculated inundation impacts for 267 countries and regions using four RCP and three SSP scenarios, and four GCMs. Because the inundation impact data from these studies are updated every ten years, linear interpolation is carried out. Then, inundation impacts are fed into the AIM/Hub model as the total amount of capital loss due to inundation. The capital loss is allocated based on the total capital of each region.

The inundation impact includes not only loss of capital but also expenditures for urgent use (e.g., relief). However, the contents of the disaster data from EM-DAT, which is the source of the inundation impact regression, are unclear as to whether both of these are included. Considering this uncertainty, we reduced the inundation impact by 20% and introduced the inundation impact into the AIM/Hub model as a sensitivity analysis.

We mainly analyzed and herein discuss household consumption changes as a representative macroeconomic indicator. Note that this research does not deal with adaptation measures in the model calculations but, in consideration of this, we compared our results with those of Tamura et al. [11], which calculated the cost-effectiveness of dike construction and reinforcement as adaptation measures. The following sections define adaptation in that way; changes in market equilibrium are not defined as adaptation.

## 3. Results

This section presents the household consumption change from the business-as-usual scenario as a CGE model result. Then, we compare the results among scenarios and compare them to the inundation impacts introduced to the CGE model. The first subsection is focused on the global macroeconomic impact, the second is focused on regional-scale impacts, and the last presents the results of the sensitivity analysis regarding the uncertainty of the historical disaster data.

### 3.1. Global Impact

Figure 4 shows the annual household consumption loss worldwide from 2010 to 2100 under each scenario. Under all RCP scenarios, the loss is large under SSP1 and small under SSP3. Under SSP1, the differences in the losses among RCP scenarios increase from 2085. In 2100, the loss reaches $159 billion under RCP8.5 (all amounts in US$), $142 billion under RCP6.0, $144 billion under RCP4.5 and $120 billion under RCP2.6. These results indicate the benefit of climate change mitigation, but even when the mitigation measures succeed, household consumption loss occurs to some extent under SSP1. Under SSP2, the household consumption loss varies from $101 billion under RCP2.6 to $114 billion under RCP8.5 in 2100. Under SSP3, the household consumption loss varies from $31 billion under RCP2.6 to $36 billion under RCP6.0 in 2100. Under SSP2 and SSP3, the differences in household consumption loss among RCP scenarios are small even in 2100. Under SSP3, because the difference in the cumulative inundation impact among RCP scenarios is as small as $17 billion, the difference in household consumption is also small. Under SSP1 and SSP2, the differences between the inundation impacts of RCP2.6 and RCP8.5 have similar values, at $36 billion and $30 billion, respectively. Therefore, taking these results as a whole, the difference in household consumption among RCP scenarios is affected not only by the different amounts of inundation impact but also by the socioeconomic assumptions. These assumptions impact on the amount of household consumption loss and the effect of climate change mitigation.

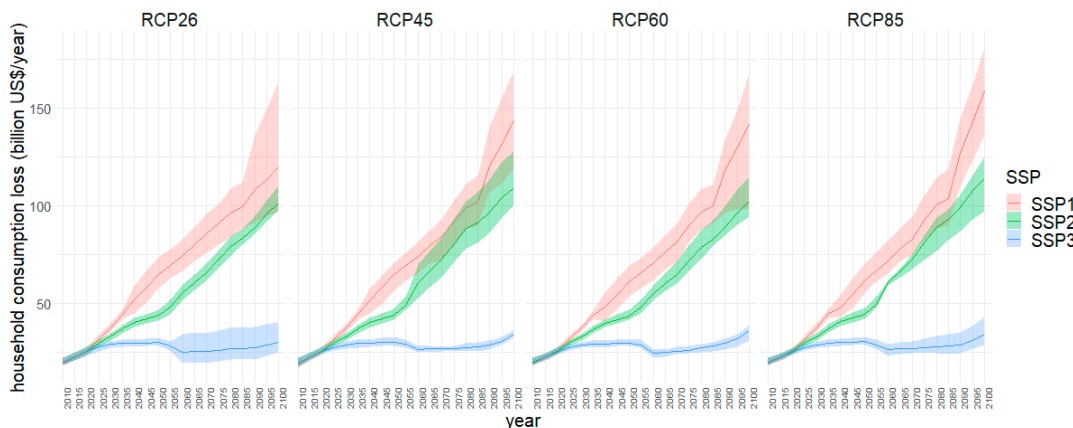

**Figure 4.** Annual household consumption loss worldwide (billion US$/year). The lines show the average value of the results by four general circulation models. The ranges show the upper and lower limits of results for these.

The sum of household consumption losses from 2005 to 2100 under RCP8.5 is $6.5 trillion under SSP1, $5.4 trillion under SSP2 and $2.6 trillion under SSP3. These loss sums are about 12 times larger than the cumulative inundation impacts. This result shows that the cumulative, macroeconomic effects of capital loss cause larger household consumption loss than the amount of direct capital loss. This result shows the importance of considering CGE.

Figure 5 shows the percentage of household consumption loss compared to household consumption worldwide in 2100. The percentages are 0.049% (RCP2.6) to 0.066% (RCP8.5) under SSP1, 0.045% (RCP2.6) to 0.050% (RCP8.5) under SSP2, and 0.027% (RCP2.6) to 0.032% (RCP8.5) under SSP3. Because the household consumption assumed in SSP1 is larger than in SSP2, and SSP2 is larger than SSP3, the difference in loss percentage among SSP scenarios is smaller than that in the loss amount, but with same the trend for the SSP scenarios. We also observed a similar trend in GDP changes.

### 3.2. Regional Impact

Figure 5 shows the percentage of household consumption loss in all 17 regions and worldwide in 2100. The proportion of household consumption change varies significantly, from a 0.18% loss (XE25 under SSP1) to a 0.04% increase (CIS under SSP1), among regions and SSP scenarios.

Focusing on each SSP scenario, under SSP1, relatively large household consumption losses of 0.18% in XE25, 0.11% in XSE and 0.13% in XSA are calculated. The inundation impact relative to the whole region's economy is much larger than the worldwide level in XSE, XSA, XME, XLM, CHN, and XOC. In XOC and CHN the loss proportion is smaller than that worldwide. In XE25 the inundation impact is small, but the loss proportion is the largest of all regions. In XER, TUR, CIS, and XNF, the household consumption increased under RCP8.5 despite a capital loss (although this was small). Under SSP2, relatively large household consumption losses of 0.12% in XSA, 0.09% in XSE, and 0.13% in XME are calculated. These regions suffered relatively large inundation impacts and, in the other regions, we did not find a clear regional trend of household consumption loss with inundation impact. Under SSP3, relatively large consumption losses of 0.16% in XSA and 0.17% in XME are calculated. In XAF, XNF, JPN, CHN, and TUR, household consumption increased despite capital loss. Because the impact of capital loss "ripples" out via its effect on the equilibrium and international trade, the household consumption in a region is affected not only by that region's inundation impact but also by the situation in other countries.

Comparing the worldwide loss proportions under RCP8.5 with RCP2.6, household consumption loss decreased in RCP2.6. This comparison indicates the benefit of climate change mitigation for SLR. On the regional scale, climate change mitigation does not always decrease household consumption

loss. Perhaps regions with increased losses under climate change mitigation have been affected by this ripple effect.

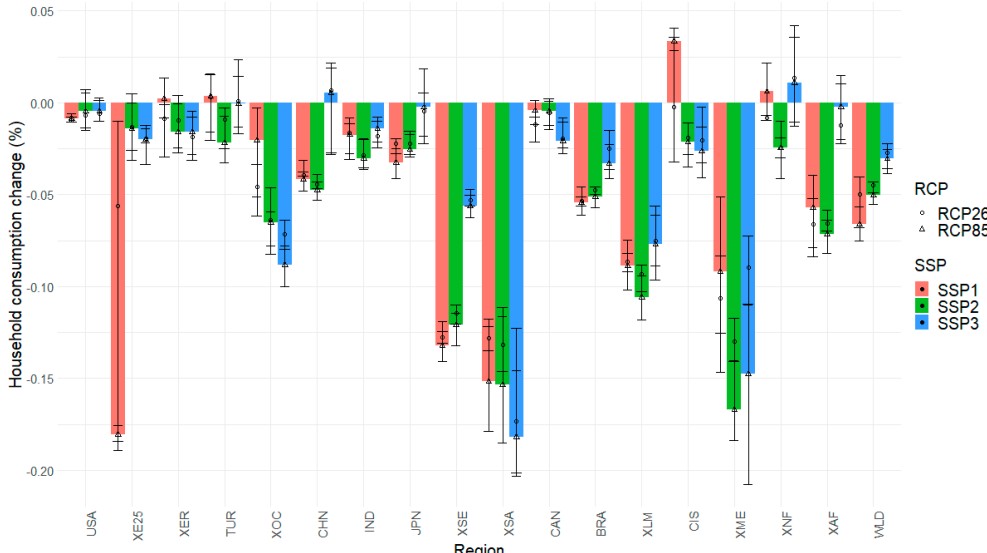

**Figure 5.** The percentage of household consumption change in 2100 in each region and worldwide (%). The bar shows the average value under representative concentration pathways (RCPs) scenario number 8.5 and the point shows the average value under each RCP scenario.

## 3.3. Sensitivity Analysis

The right side in Figure 6 shows the result when 80% of the inundation impact is introduced to the CGE model as a capital loss. In this analysis, the response varies from scenario to scenario. Under some scenarios, the household consumption loss increased despite the decreased capital loss, however, the trend among SSP scenarios is consistent under all RCP scenarios. Under SSP1, the difference in household consumption loss between RCP2.6 and RCP8.5 is $18 billion (0.0077%). Under SSP2 and SSP3, the differences are $10 billion (0.0042%) and $4 billion (0.0035%), respectively. This result shows that the socioeconomic assumptions affect the household consumption loss and the impact of climate change mitigation when the inundation impact is reduced by 20%; the effects have the same trends as under the main scenario.

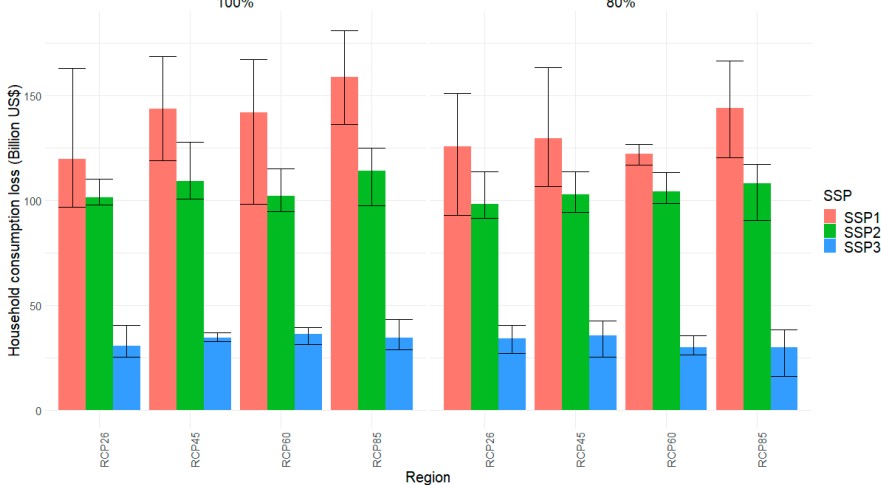

**Figure 6.** The worldwide household consumption loss in 2100 (billion US$) The right side of the figure shows the result with an inundation impact reduced by 20%.

## 4. Discussion

In this section, based on the above-mentioned results, we discuss five points about the economic impacts of SLR.

The first point is about the magnitude of the impact on the world economy. In this study, we have found that the worldwide household consumption loss is from 0.027% up to 0.066% and that the largest loss proportion happens under RCP8.5 with SSP1. The household consumption loss due to inundation impact, compared to the worldwide economy, is relatively small. However, we cannot always therefore judge that the impact is not serious for a coastal area, because the household consumption loss may concentrate in such an area.

Second, we discuss the socioeconomic factors that cause the differences in the household consumption loss amount and proportion. Comparing SSP scenarios, larger household consumption losses are calculated under higher GDP-growth scenarios (SSP1 > SSP2 > SSP3). One of the reasons for this is that the economic growth in a coastal area is larger than that of other areas, especially in higher-growth scenarios, and increases the assets exposed to inundation caused by SLR. Another possible reason can be found when comparing the impact of climate change mitigation among SSP scenarios, which relates to a limitation of this study: its lack of an explicit vulnerability and adaptation capacity, beyond population and GDP. Under SSP2 and SSP3, we find a small difference in the consumption loss among RCPs, while SSP1 shows a larger difference in relation to the others. Because SSP1 and SSP2 have similar differences in inundation impact among the RCP scenarios, the difference in inundation impact alone cannot be the cause of the difference in the effect of mitigation among the SSP scenarios. Another possible reason is technological development, which can increase production per unit of capital and therefore causes capital loss to have a greater effect on household consumption. We conducted a sensitivity analysis to examine an uncertainty of the historical disaster data. In this analysis, we confirmed the robustness of the impact of the socioeconomic assumptions on the household consumption loss and the effect of climate change mitigation. However, household consumption loss occurs to some extent even when mitigation measures succeed. This result indicates the need to evaluate more stringent climate change mitigation scenarios, which are consistent with the 1.5 °C target.

Third, we focus on the regional-scale impacts. A large inundation impact in a region does not always cause large household consumption loss in that region due to the trade network. Regions that export capital-intensive goods are susceptible to inundation impact and can increase their income by increasing prices and production levels. Oppositely, regions that import such goods are adversely affected by rising prices. It can be considered that the effects of inundation impact spread internationally due to trade.

Fourth, we compare our result with other related research to clarify the target of our assessment. Regarding comparison with recent, relevant studies, Schinko et al. [8] conducted a multiple-model assessment of the macroeconomic impacts of coastal flooding due to SLR based on two RCP scenarios (RCP4.5 and RCP2.6). Their results show that global economic losses can amount to more than 3% under RCP2.6 and more than 4% under RCP4.5 without further adaptation and assuming high ice-melting, which exceed our results. This is because they consider not only SLR but also coastal floods, including extreme inundation. Our research accounts only for inundation due to SLR and tidal change and, thus, constant inundation. Also note that this study does not address non-market value loss. By contrast, Sao et al. [26] addressed the loss of recreational value due to beach erosion. Their research estimated that the amount of damage to sand beaches in Japan due to climate change will be about $500 million per year (2081 to 2100) under RCP8.5. The result of our research shows that household consumption loss in Japan would be about $1.4 billion in 2100 under RCP8.5 with SSP2. Sao et al. [26] calculate the loss of consumer surplus in the transportation market caused by the decline in traffic demand due to sand beach erosion as a loss of non-market value. Therefore, it is difficult to make a simple comparison between the loss of household consumption shown by our study and the loss of non-market value by Sao et al. [26]. However, since the loss of non-market value by Sao et al. [26], which amounts to about 36 percent of the loss of household consumption in this study, is additional to the loss of market

value, the impact of SLR on the overall economy would be significantly larger than the results of the calculations in this study. These results show the importance of considering not only market value, such as capital loss, but also non-market value when assessing the impact of SLR on the economy as a whole.

Fifth, we discuss the effect of adaptation measures. Tamura et al. [11] researched the cost-effectiveness of building and reinforcing dikes against SLR due to climate change. In their study, the effect was calculated as a reduction in the inundation impact, and the cost was calculated as the cost of building, reinforcing, and maintaining the dikes. They estimated that the cost of dikes in RCP8.5 will be approximately $930 million annually between 2090 and 2100. The differences in the inundation impact between when the dikes are reinforced and when they are not were calculated to be approximately $5.1 billion per year under SSP1 and $3.7 billion per year under SSP3. Therefore, the cost-effectiveness of adaptation measures to reinforce the dikes is shown to have a value greater than one. They calculated the reduction in the inundation impact as an effect. However, considering our results, if the household consumption change is dealt with as an effect, the cost-effectiveness should be further improved.

## 5. Conclusions

This research estimated household consumption change as a welfare measure caused by SLR due to global climate change covering the entire 21st century. We used the AIM/Hub model, a global computable general equilibrium model, to calculate not only direct inundation impact but also the general equilibrium and cumulative effect of capital loss over time. Then, we evaluated the macroeconomic impact on 17 regions around the world. To understand the future uncertainties comprehensively, we used multiple socioeconomic assumptions, climate change mitigation levels, and climate models. This method allows us to consider differences in future socioeconomic assumptions and climate change mitigation measures.

The global household consumption loss thus calculated is 0.049% (RCP2.6) to 0.066% (RCP8.5) under SSP1, 0.045% (RCP2.6) to 0.050% (RCP8.5) under SSP2, and 0.027% (RCP2.6) to 0.032% (RCP8.5) under SSP3. The household consumption loss was calculated to be about 12 times the direct inundation impact because of its macroeconomic effect. This indicates the importance of CGE calculation when assessing economic impact and adaptation measures. Comparisons among the SSP and RCP scenarios indicate that socioeconomic assumptions affect the amount of household consumption losses and the impacts of mitigation. However, the overall range of macroeconomic implications associated with different RCP scenarios is small relative to the total household consumption losses, even in the socioeconomic assumption where the mitigation measures are most effective. This result would mean that even if mitigation measures are successful, household consumption change would occur to some extent, indicating the importance of adaptation. It also implies that 2 °C climate stabilization, which is often considered as RCP2.6, might not be sufficient to prevent climate change damage and it would be better to assess more stringent climate change mitigation.

The regions suffering from relatively large household consumption losses are the EU, East and Southeast Asia, and Asia under SSP1, Southeast Asia, Asia, and the Middle East under SSP2, and Asia and the Middle East under SSP3. This is due to their relatively large inundation direct damage. However, importantly, household consumption loss in the EU under the SSP1 scenario is high despite a small direct inundation impact. Additionally, there are regions where household consumption losses are not in line with those regions' direct impacts. The interlinkage effects through trade, including price increases and replacing the production of capital-intensive goods, seemed to cause such consequences.

We divided the world into 17 regions and identified the regions that suffered relatively large household consumption losses due to SLR. It is possible that larger losses occur on the scale of individual countries. However, it is difficult to evaluate country-scale impacts using the global-scale model used in this study. To deal with a country-scale impact, it is necessary to conduct an evaluation using a national-scale model. The results of this study, such as the affected region and the socio-economic

factors that worsen household consumption loss, are useful for selecting and evaluating target countries. Therefore, improvements to our research can be summarized by means of two points: first, a more stringent climate change mitigation scenario, consistent with the 1.5 °C target, should be assessed, and second, a more detailed regional- or national-scale study should aim to identify the most serious impacts.

In this study, we assessed the global economic impact of inundation due to SLR, taking into account the uncertainties of the future social situation and climate condition. The main findings were that the global household consumption loss proportion was 0.045%, in the range 0.027−0.066%, and the magnitude of this household consumption loss and the impact of climate change mitigation relied on the socioeconomic assumptions.

**Author Contributions:** Conceptualization, O.N.; methodology, O.N.; software, S.F.; validation, O.N.; resources, M.T.; writing—original draft preparation, O.N.; writing—review and editing, S.F., M.T., K.T., J.T., and Y.H.; visualization, O.N.; supervision, S.F.; project administration, O.N.; and funding acquisition, S.F. All authors have read and agreed to the published version of the manuscript.

**Funding:** This study was supported by the Environment Research and Technology Development Fund (S-14, 2-1908 and 2-2002) provided by the Environmental Restoration and Conservation Agency, Japan.

**Acknowledgments:** The authors acknowledge that the inundation data were provided by the laboratory of Hiromune Yokoki, Ibaraki University.

**Conflicts of Interest:** The funders had no role in the design of the study; in the collection, analysis, or interpretation of data; in the writing of the manuscript, or in the decision to publish the results.

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
