# Peer review of "An Assessment of Global Macroeconomic Impacts Caused by Sea Level Rise Using the Framework of Shared Socioeconomic Pathways and Representative Concentration Pathways"

_sustainability, doi:10.3390/su12093737_

Round 1
Reviewer 1 Report
- In table 1. the following regions are included twice: USA, EU, Rest of EU. Please correct
- Given that non-market value loss (e.g. landscape, recreational, etc.) are not estimated in the current study how comparable are the results with similar studies (for instance Sao et al.)? The figures provided in lines 291-297 do not explain enough the diference in estimations.
Author Response
Response to Reviewer 1 Comments
Point 1: In table 1. the following regions are included twice: USA, EU, Rest of EU. Please correct
Response 1: Thank you for pointing this out. We are pointed out by another reviewer that table 1 is not informative and it should be directly on the text. Therefore, we described the information in lines 100-104.
“The target region is the whole world, divided into following 17 regions clustered by geography and economy, USA (USA), EU (XE25), Rest of EU (XER), Turkey (TUR), Australia and New Zealand (XOC), China (CHN), India (IND), Japan (JPN), Rest of East and South East Asia (XSE), Rest of Asia (XSA), Canada (CAN), Brazil (BRA), Rest of South America (XLM), Former USSR (CIS), Middle East (XME), North Africa (XNF), and Sub-Sahara (XAF).”
Point 2: Given that non-market value loss (e.g. landscape, recreational, etc.) are not estimated in the current study how comparable are the results with similar studies (for instance Sao et al.)? The figures provided in lines 291-297 do not explain enough the diference in estimations.
Response 2: Thank you very much for your comment. There was a lack of explanation of the non-market value in Sao et al. We added the following explanation in lines 327-333.
“Sao et al. [26] calculates the loss of consumer surplus in the transportation market caused by the decline in traffic demand due to sand beach erosion as a loss of non-market value. Therefore, it is difficult to make a simple comparison between the loss of household consumption shown by our study and the loss of non-market value by Sao et al. [26]. However, since the loss of non-market value by Sao et al. [26], which amounts to about 36 percent of the loss of household consumption in this study, is additional to the loss of market value, the impact of SLR on the overall economy would be significantly larger than the results of the calculations in this study.”
Reviewer 2 Report
The paper aims at analysing the global economic impact of inundation due to sea level rise by using general equilibrium model. The topic of the paper is interesting. The article is well structured, however, additional work is required. In fact, in this present form, it is a bit hard to read for those are not very familiar with these models. Therefore, my overall assessment of the document is barely sufficient. In the following, my comments:
P3_L102-104: Authors wrote “…MIROC-ESM, GFDL-102 ESM2M, NorESM1-M, and IPSL-CM5A) based on four RCP scenarios (RCP2.6, RCP4.5, RCP6.0, and 103 RCP8.5), and the economic impact is calculated based on three SSP scenario (SSP1, SSP2, SSP3)” Even if there is a reference, I think that more explication are needed. I suggest adding, at least, a footnote for the four global circulation models as well as a description for all the scenarios.
P3_L121: Table 1 is not informative. Maybe, you can replace it and put the list of the countries directly on the text with acronyms in brackets. In addition, why countries are grouped in such a way?
SECTION 2.1. Overview: I suggest adding the complete list of variables used in the models as well as some descriptive statistics. The list of the variables should also include a precise definition of each of them and the source.
SECTION 2.3. CGE model calculation: A more detailed description of the models is required.
Author Response
Response to Reviewer 2 Comments
The paper aims at analysing the global economic impact of inundation due to sea level rise by using general equilibrium model. The topic of the paper is interesting. The article is well structured, however, additional work is required. In fact, in this present form, it is a bit hard to read for those are not very familiar with these models. Therefore, my overall assessment of the document is barely sufficient. In the following, my comments:
Thank you very much for your positive assessment and helpful comments. We will respond to each comment in detail below.
Point 1: P3_L102-104: Authors wrote “…MIROC-ESM, GFDL-102 ESM2M, NorESM1-M, and IPSL-CM5A) based on four RCP scenarios (RCP2.6, RCP4.5, RCP6.0, and 103 RCP8.5), and the economic impact is calculated based on three SSP scenario (SSP1, SSP2, SSP3)” Even if there is a reference, I think that more explication are needed. I suggest adding, at least, a footnote for the four global circulation models as well as a description for all the scenarios.
Response 1: Thank you very much for your comment. We added the references of four GCMs and descriptions for RCP scenarios and SSP scenarios with some related values in lines 104-126.
“To understand the uncertainties comprehensively of the global circulation models (GCMs) and future climate change mitigation policies, the SLR is calculated using four GCMs (MIROC-ESM[12], GFDL-ESM2M[13], NorESM1-M[14], and IPSL-CM5A[15]) based on four RCP scenarios (RCP2.6, RCP4.5, RCP6.0, and RCP8.5)[16], and the economic impact is calculated based on three SSP scenario (SSP1, SSP2, SSP3) [17]. The RCP scenarios are the GHG concentration trajectory based on the future radiative forcing, and they describe different climate futures. The four scenarios used in this study means that the future radiative forcing reach to 2.6w/m2, 4.5w/m2, 6.0w/m2, 8.5w/m2 in 2100, respectively. The IPCC’s Fifth Assessment Report WG1 reports that the average temperature increase over the late-21st century (2081-2100) compared to the early-21st century (1986-2005) reaches 1.0°C (0.3-1.7°C) under RCP 2.6, 1.8°C (1.1-2.6°C) under RCP 4.5, 2.2°C (1.4-3.1°C) under RCP 6.0, and 3.7°C (2.6-4.8°C) under RCP 8.5. The SSP scenarios are based on two dimensions, the level of challenges to adaptation and that of mitigation. SSP1, SSP2, and SSP3 used in this study assume a "Sustainability" scenario that is easy to adapt and mitigate, a "Middle of the Road" scenario that continues the current trend, and a "Regional Rivalry" scenario that is difficult to adapt and mitigate, respectively. Under SSP2, the world population reach 8.9 billion, and the world GDP reach $290 trillion in 2100. SSP1 assumes a society in which population growth stops (6.8 billion) because of the assumption that emission reductions are easy to achieve. In addition, a society with a large GDP ($310 trillion) is assumed because of the assumption that it is easy to pay the costs of adaptation measures. On the contrary, SSP3 assumes a society with a growing population (12 billion) and relatively small GDP growth ($150 trillion). The combination of RCP and SSP scenarios allows the assessment of climate change impacts to take into account the uncertainty of the future socioeconomic condition and the stringency of mitigation policy.”
Point 2: P3_L121: Table 1 is not informative. Maybe, you can replace it and put the list of the countries directly on the text with acronyms in brackets. In addition, why countries are grouped in such a way?
Response 2: Thank you for pointing this out. We replaced Table 1 and added region names and the explanation of the way to group in lines 100-104.
“The target region is the whole world, divided into following 17 regions grouped by geography and economy, USA (USA), EU (XE25), Rest of EU (XER), Turkey (TUR), Australia and New Zealand (XOC), China (CHN), India (IND), Japan (JPN), Rest of East and South East Asia (XSE), Rest of Asia (XSA), Canada (CAN), Brazil (BRA), Rest of South America (XLM), Former USSR (CIS), Middle East (XME), North Africa (XNF), and Sub-Sahara (XAF).”
Point 3: SECTION 2.1. Overview: I suggest adding the complete list of variables used in the models as well as some descriptive statistics. The list of the variables should also include a precise definition of each of them and the source.
Response 3: Thank you very much for your comment. We added the statistics to help understanding the scenarios used in this study in lines 111-124 and added the table about the data used in each process. The additional table is shown below.
Process |
Input data |
Source |
1) SLR calculation |
GHG concentration |
RCP scenarios [16] |
2) Inundation area calculation |
Sea level Land elevation High tide water level |
1) ETOPO1 [19] TPXO7.2 [20] |
3) Multiple regression analysis |
Historical disaster damage GDP, Population |
EM-DAT [18] World bank’s statistics [21] |
4) Inundation impact calculation |
Inundation Area GDP, Population |
2) SSP scenarios [17] |
5) CGE calculation |
Inundation impact Socioeconomic conditions |
4) SSP scenarios [17] |
6) Economic analysis |
Household consumption |
5) |
Point 4: SECTION 2.3. CGE model calculation: A more detailed description of the models is required.
Response 4: Thank you very much for your comment. We added the detailed description in lines 176-183.
“The production sectors maximize profits under multi-nested constant elasticity substitution (CES) functions and individual input prices. Household expenditures on each commodity are described by a linear expenditure system (LES) function. The parameters adopted in the LES function are recursively updated in accordance with income elasticity assumptions. The saving ratio is endogenously determined to balance saving and investment, and capital formation for each good is determined by a fixed coefficient. The Armington assumption is used for trade (CES and constant elasticity of transformation function is used), and the current account is assumed to be balanced.”